# What does it take to design digitally enabled performance management and incentive interventions for community health programs: Lessons from Ethiopia

Alemnesh Hailemariam Mirkuzie[1,2,3]◉*, Yared Kifle[1]◉, Gizachew Tadele Tiruneh[1]◉, Girma Tadesse[1‡], Getnet Alem Teklu[1‡], Esubalew Sebsibe[1‡], Eyoel Mitiku[4‡], Aklilu Abera[5‡], Wondwossen Shiferaw[5], Birhutesfa Bekele[1‡], Wuleta Aklilu Betemariam[6‡], Desalew Emaway[1‡]

**1** John Snow Inc Training and Research (JSI), Addis Ababa, Ethiopia, **2** Jimma University, Jimma, Ethiopia, **3** University of Washington, Institute for Health Metrics and Evaluation, Seattle, Washington, United States of America, **4** Dimagi, Addis Ababa, Ethiopia, **5** LivingGoods, Addis Ababa, Ethiopia, **6** JSI, Washington District of Colombia, Arlington, Virginia, United States of America

◉ These authors contributed equally to this work.
‡ AA, BB, DE, EM, ES, GAT, GT,WAB and WS also contributed equally to this work.
* Alemnesh_hailemariam@et.jsi.com

## Abstract

The Health Extension Program (HEP) in Ethiopia has faced multifaceted challenges, including declining motivation and suboptimal performance of Health Extension Workers (HEWs). These issues have significantly impacted Reproductive, Maternal, Newborn, and Child Health (RMNCH) outcomes. To address these gaps, JSI, in collaboration with key partners, designed digitally enabled performance management (PM) and performance-based incentive (PBI) interventions integrated into the electronic Community Health Information System (eCHIS). A multi-approach design process was implemented, including a landscape review of existing strategies, human-centered design (HCD), and participatory co-design workshops. National and regional stakeholders contributed to the development process to ensure contextual relevance. A hybrid framework combining Management by Objectives (MBO) and the DESC (Digitally enabled, Equipped, Supervised, Compensated) model guided the design. The digitally enabled PM/PBI interventions required significant advancement to the eCHIS application suite, such as enhancing the existing focal person application (FPA) with real-time monitoring dashboards, digital target setting, and automated supervision features, and developing a national eCHIS dashboard for supervisory support, data-informed performance evaluation, and decision making. Twenty-two key performance indicators (KPIs) were identified to measure outputs, health outcomes, and supervisory processes. The intervention integrated digitally supported supervision and mentorship to drive performance improvements. Stakeholders proposed incentivizing the HEWs, supervisors, and HPs who record high performance biannually as a team and/or an individual with non-financial

**Data availability statement:** Data underlying the study findings are included as Supporting information file.

**Funding:** The study received funding from the Child Investment Fund Foundation, grant number 2102-06118 (previously 2010-05155). DE received the award for the eCHIS project and embedded studies. The URL of CIFF's website is https://ciff.org/. The funders had no role in study design, data collection and analysis, decision to publish, or preparation of the manuscript.

**Competing interests:** The authors have declared that no competing interests exist.

or mixed incentives. In conclusion, the participatory design process resulted in robust, scalable PM/PBI interventions tailored to Ethiopia's HEP. Digitally enabled tools, when aligned with supportive supervision and sustainable incentive strategies, have the potential to improve HEW motivation, RMNCH outcomes, and health system accountability. This model offers valuable lessons for other low-resource settings implementing performance management systems in community health programs.

## Author summary

In designing the performance management interventions for improving Ethiopia's community health program, we integrated digital tools and performance-based incentives to enhance community health workers' motivation, performance, and service delivery. The interventions combined digital innovations, i.e., the supervisor application enhanced with Target setting, Supervisor task manager, and Dashboard functionalities; electronic Community Health Information System, and national dashboard with incentives tied to performance metrics. Leveraging existing systems and resources, these tools enabled real-time tracking of health services, streamlined supervision, and improved decision-making. By employing a participatory design approach, we engaged stakeholders to ensure the design was practical, culturally relevant, and aligned with national health goals. We made special considerations in selecting and communicating indicators for measuring targeted performance, incentive types, frequency, and recipients tailored to local needs and contexts. The intervention design addressed systemic challenges while fostering ownership and scalability by involving community health workers, supervisors, and decision-makers. This participatory approach offers a sustainable model for improving health outcomes in similar contexts. Addressing infrastructure gaps, ongoing technical support, and testing these interventions to maximize their impact are important considerations. Testing the designed interventions to evaluate their effect on community health workers' motivation, performance, and health outcomes is the next project's activity.

## Introduction

Achieving Universal Health Coverage and Sustainable Development Goal 3 necessitates a motivated and capable health workforce and harnessing the power of digital technologies [1–3]. Ethiopia's Health Extension Program (HEP), established in 2003, serves as a cornerstone of expanding primary health care, particularly in rural and underserved communities [3,4]. However, despite its significant contributions to expanding access to essential services, in recent years, the HEP has faced persistent challenges, including low workforce motivation, performance, and high turnover, leading to gaps in achieving optimal reproductive, maternal, newborn, and child health (RMNCH) outcomes [5]. To address these challenges, the Ethiopian Ministry

of Health (MoH) developed the HEP optimization roadmap, which emphasizes the need for innovative performance management systems to enhance service delivery, health extension workers' (HEWs) motivation, and efficiency [6].

By implementing the electronic Community Health Information System (eCHIS) since 2018, the Ethiopian HEP has digitally transformed community health service provision and health information systems. This sets the foundation for designing integrated innovative digital performance management interventions. Studies demonstrated the benefits of digital performance management systems in counteracting dysfunctional systems and digital tools and frameworks that streamline supervision, monitoring, and incentivization processes, thereby addressing systemic inefficiencies and fostering accountability [3,7–10].

Among the well-known performance management frameworks relevant to improving community health care workers' performance and motivation are Management by Objective (MBO) and the Digitized, Equipped, Supervised, and Compensated (DESC) frameworks [11]. The MBO aligns organizational goals with individual responsibilities, promoting goal-oriented activities and transparency in health service delivery [12]. The DESC framework prioritizes operational functions such as supervision, provision of digital tools, and appropriate compensation to improve performance [11]. These frameworks have been successfully implemented in other contexts, including community health programs in Kenya and Uganda, delivering commendable results [11–13] and lending great lessons for Ethiopia's HEP. The community health program in Ethiopia, despite being well-resourced, government-led, and mature, its performance management systems remain dysfunctional lacking a clear guiding framework, contributing to the poor performance of the program [5].

The use of data and digital tools has emerged as a transformative strategy in health system strengthening [8–10,14]. Digital tools such as the electronic Community Health Information System (eCHIS) application suite enable population registration, service provision, real-time data collection, and analysis, facilitating data informed decision-making and timely corrective actions [15].

The Ethiopia eCHIS application suite integrates the eCHIS HEW application, referral application, focal person application, and eCHIS dashboard. The eCHIS suite can serve as the foundation for digitally enabled performance management innovations, providing HEWs, supervisors, and program managers with actionable insights to enhance service delivery. By integrating data-driven approaches, the HEP can overcome traditional barriers to performance monitoring and improve health outcomes. Supervision and incentivization are critical components of effective performance management systems. Evidence from studies underscores the importance of regular supervision, digital supervision, personalized dashboard, and performance-based incentives in motivating CHWs and improving service quality [15–17]. In Ethiopia, the introduction of performance-based incentives, both financial and non-financial, aims to address existing gaps in HEWs' motivation, retention, and performance. Studies underscore important enablers such as IT infrastructure foundation, systems, and governance structure that should be considered when designing digital health and performance management interventions [3,8–10,14,16,18–20]. According to a scoping review, designing and testing digital health interventions require active engagement of stakeholders, aligning interventions to existing workflow and government priorities, leveraging existing systems, and iterative learning to ensure adoption, feasibility, sustainability, and scalability of the interventions [10].

Given the context-specific challenges faced by Ethiopia's HEP, this paper describes the design of a digitally enabled PM/PBI system that leverages the eCHIS application suite. Informed by global evidence, participatory approaches, and local insights, we designed performance management interventions to improve HEW performance, strengthen supervision, and achieve better RMNCH outcomes. This work contributes to the growing body of evidence on digital solutions for community health systems in low-resource settings.

## Methods

### Setting and context

Ethiopia's healthcare system comprises three tiers: primary, secondary, and tertiary levels. At the primary level, there are health posts, which serve as the first point of care and are staffed by HEWs who deliver HEP services to an estimated population of 5,000. There are more than 17,500 health posts across the country, and approximately 40,000 HEWs who

provide HP services [5]. The HEP uses an integrated community health information system [5]. Aligned with the country's information revolution agenda, the system has been digitized and scaled up to HPs across the country since 2018 [5,21–23]. The eCHIS application suite allows HEWs to register households and population, provide health services, for referral and receive feedback, and generate reports for performance monitoring and monthly reporting.

JSI is one of the partners supporting eCHIS scale-up for improving RMNCH service provision through funding from the Children Investment Fund Foundation (CIFF). It is a five-year project from 2020-2025 intending to i) implement eCHIS to improve care quality, referral linkages, and data use for evidence-based decision-making at all levels; ii) leverage eCHIS to transform community-level health data for HEW performance management and targeting primary health services and iii) design, test, and implement performance management, performance-based incentives, and biometric innovations on the eCHIS application suite for national scale-up of an optimized HEP.

In March 2021, JSI and the MoH launched the performance management initiative of the project in the presence of the MoH leadership, CIFF, and development partners working in the space to create a shared understanding and foster ownership of the initiative. The initiative has two major interventions, i.e., digitally enabled performance management (PM) and digitally enabled performance-based incentives (PBI). Except for the incentivization, the PM and PBI interventions are the same and hence hereafter referred to as PM/PBI

In consultation with relevant MoH and regional stakeholders, we developed criteria for selecting three woredas for PM and three for PBI to make the design evidence-informed and context-appropriate. The criteria were based on health indicators, operational ease, security, and ICT infrastructure (tablet, internet connectivity, and power source for charging the tablet). Accordingly, Awabel woreda from the Amhara region, Gimbichu from the Oromia region, and Mirab Abaya from the South Ethiopia region were selected for PBI. Dembecha woreda from the Amhara region, Lume from the Oromia region, and Lemo woreda from the Central Ethiopia region (former SNNP region) were selected for PM interventions. The initiative has been implemented in partnership with the MoH, Dimagi (a technical partner), and Living Goods, which provided comprehensive expertise in designing the PM/PBI interventions.

## Design approaches

The PM/PBI systems were designed through a multi-step, iterative process that combined evidence-based approaches, stakeholder engagement, and technology-driven innovations. The key design steps included a landscape review, human-centered design (HCD), participatory co-design workshops, leveraging existing resources, and harnessing the eCHIS application suite.

**Landscape Review:** The intervention began with an extensive review of existing policies, strategies, and practices within the Ethiopian health system. In total, 63 documents, including national health policies, global practices, and existing performance management systems, were reviewed. This was complemented with 31 exploratory qualitative in-depth interviews and three focus group discussions with key informants. The informants came from regional health bureaus, zonal health departments, woreda health offices, health centers, and health posts in Amhara, Oromia, and Southern Ethiopia regions. These qualitative data were analyzed using Atlas. Version 8 software. Through the qualitative methods, existing practices and gaps in performance management systems, particularly supervision, data use, and incentives, were explored.

**Human-centered design (HCD)** is the other technique employed to gather insights into developing context-appropriate PM/PBI digital interventions embedded in the eCHIS application suite. Insights were gathered from the HEWs, HC supervisors, and woreda HEP coordinators. Six key users from the Amhara region, 12 from Oromia, and 11 from the Central Ethiopia region participated in the HCD insight gathering for 2.5 weeks. In total, 10 HEWs, six supervisors, five maternal and newborn health care providers, three eCHIS coordinators (one from each woreda), three HEP coordinators (one from each woreda), and three HR managers (one from each woreda) participated in the HCD interviews. Participatory interview techniques were used to dig deep into the user experience, build empathy with users, and understand users' words, actions, thoughts, and feelings. This helped to define specific opportunities for design improvements to the existing supervisor application, hereafter referred to as the Focal Person Application (FPA), to address users' needs tailored to their

contexts. The factors that influence app use, quality of supervision, data-informed decision-making, and quality of data entry were also explored. Furthermore, insights on the type of PBI, the frequency of handing out the incentive, and who to be involved in the performance verification were gathered.

**Stakeholder engagement and co-design Workshops:** Engaging stakeholders across all levels of the health system was central to the intervention's design. National and sub-national working groups and implementing partners participated in the co-design workshops. Six co-design workshops were conducted with 320 stakeholders from HPs, HCs, and regional health offices to review and validate the findings and recommendations from the landscape review and HCD, to shape the thinking around the design and implementation of proposed PM/PBI interventions.

### Design frameworks

The design of PM/PBI interventions was guided by a combination of Management by Objective (MBO) and Digitized, Equipped, Supervised, and Compensated (DESC) frameworks. MBO ensured alignment of performance indicators with strategic goals, while the DESC model addressed operational aspects, including digital enablement, equipping HEWs, supervision, and compensation [11,12].

1. *Digitally Enabled*: Enhancing the eCHIS application suite to enable real-time performance monitoring and supervision as well as developing a national eCHIS dashboard for performance tracking, evaluation, and data-informed decision-making.

2. *Equipped*: Ensuring HEWs and supervisors had access to essential tools, resources, training, essential RMNCH medicines, and supplies.

3. *Supervised*: Strengthening supervision through digital tools and mentorship support.

4. *Compensated*: Providing incentives to motivate HEWs, supervisors, and health posts.

**Digital Tool Development:** Enhancing the FPA and developing the eCHIS dashboard were key undertakings on digital tools/solutions. The FPA was enhanced with key performance management features informed by the evidence from the landscape review and HCD. Dimagi, the technology partner, enhanced the existing FPA based on the requirement documents produced by JSI, MOH, and LivingGoods. The enhancement followed an iterative process complemented by user acceptance testing. Similarly, based on the requirement documents developed by JSI and LivingGoods, Dimagi developed a national eCHIS dashboard.

### Ethical considerations

Ethical clearance was obtained from the Ethiopian Public Health Association (EPHA) Research Ethics Review Board dated June 30, 2022, with reference # EPHA/OG/886/22. Informed oral consent was obtained from individuals who participated in the consultation, HCD, and co-design workshops.

## Results

### Key findings of the landscape review and HCD

The landscape review identified challenges within the Ethiopian HEP, such as a lack of standardized performance management tools, weak supervision systems, and the absence of clear performance-based incentives. The review recommended leveraging Ethiopia's eCHIS application suite to facilitate digital performance monitoring and enable data-driven decision-making. The review highlighted the need for digitized solutions to enhance the efficiency and accountability of HEWs and their supervisors. It also emphasized the need for non-monetary incentives tailored to Ethiopia's economic and cultural context to ensure sustainability.

The human-centered design (HCD) approach was integral to shaping the intervention. Insights gathered from HEWs, supervisors, and decision-makers identified key pain points, including inconsistent supervision, limited access to actionable data, and inadequate incentivization. These insights guided the creation of user-centric tools that addressed local needs. The sessions revealed key barriers to HEW performance, including a lack of technical capacity, difficulties in navigating the eCHIS app, and inconsistent supervision and support. Supervisors highlighted challenges in monitoring HEW performance using paper-based tools, which were labor-intensive and prone to errors. HCD findings informed the development of requirement documents to enhance the FPA, including a streamlined interface for supervisors, real-time performance dashboards, and automated supervision tools. Participants also expressed a preference for non-monetary incentives, such as career advancement opportunities, certificates of recognition, and in-kind rewards.

**Key findings from co-design workshops and stakeholder engagement**

This participatory design process engaged stakeholders from various levels of the health system. Stakeholder engagement was pivotal in shaping the intervention's design. Through national and regional consultations and deliberations, the PM/PBI technical working groups derived KPIs from the goals of the HEP and RMNCH to ensure the intervention's relevance and effectiveness. Relevant global evidence and national strategies were reviewed, taking into account the scope of the eCHIS application. Moreover, the selected indicators must be attributable to the tasks of the HEWs and supervisors to have their direct influence over the indicators and should be among the MoH's health information management system indicators. The KPIs must be measurable, verifiable, and understandable to the HEWs and supervisors to improve their performance. In total, 22 KPIs were selected. Of these, six were impact indicators to measure HP performance; 10 were output and outcome indicators to measure HEW performance, and six were output and outcome indicators to measure supervisors' performance. For the list of KPIs, see S1 Table. The selected indicators provide measurable benchmarks for assessing HEWs, supervisors, and health post performance. By monitoring KPIs through the enhanced FPA and eCHIS dashboard, supervisors, performance verification teams, and managers can identify high-performing individuals, teams, and facilities to recognize their contributions.

Through co-design workshops, stakeholders, including HEWs, supervisors, and managers, ensured alignment of the interventions with local needs and national health priorities, enhancing the practicality, scalability, and shared understanding of the interventions. The workshops validated HCD findings, defined performance targets, and collaboratively validated the key performance indicators (KPIs) identified for monitoring HEWs and supervisors' performance and HP level outcomes. Participants emphasized the importance of aligning KPIs with RMNCH priorities and ensuring that performance verification processes are objective and transparent. They also contributed to defining performance standards and evaluation metrics, pinpointing potential implementation challenges. Workshop participants identified implementation strategies, such as biannual performance verification and incentivization, targeted training, and regular performance reviews at the health center and woreda levels. The stakeholders defined performance evaluation criteria aligning with the HEP model woreda categorization, i.e., **High Performers**: KPI scores above 85%; **Medium Performers**: KPI scores between 70–85%; and **Low Performers**: KPI scores below 70%. Stakeholders prioritized sustainable incentive mechanisms, with a focus on non-monetary rewards.

**PM/PBI interventions and implementation strategies**

1. **Digitally enabled**: Creating a digitized environment has been one of the interventions in line with the digitally enabled component of the DESC framework. This includes maximizing the use of the eCHIS application by the HEWs, enhancing the FPA, and developing an eCHIS dashboard.

   ***Enhancing the Focal Person Application (FPA):*** The performance management interventions were designed to be fully integrated with the eCHIS application suite for improved HEW performance monitoring, supervision, and accountability in the HEP. This includes enhancing the existing FPA with a user-friendly digital interface that targets supervisors to facilitate

supervision and performance management, as shown in Fig 1. Enhancing the FPA was a pivotal innovation in the intervention, enabling supervisors to automate performance tracking and streamline management processes. The FPA enabled real-time task management, automated reporting, and direct supervision of HEWs' activities. It introduced automation in target setting, supervisory and mentorship checklists, and real-time dashboards to streamline HEWs, supervisors, and HPs' activities.

1. *Plan Setting Manager:* Supervisors used this feature to set clear, measurable performance targets for HEWs based on woreda priorities, population size, and HP capacity, fostering a shared commitment to RMNCH and HEP services. The automated target-setting feature allowed supervisors to establish clear and measurable performance goals by engaging the HEWs, ensuring that the HEWs worked toward prioritized KPIs and health outcomes, such as antenatal care coverage, immunizations, and family planning.

2. *Supervisor Task Manager:* The digitized supervision tools supported mentors and supervisors in conducting systematic, structured oversight of HEW activities. Supervisors use automated checklists to identify performance gaps, address immediate needs, and provide customized support. Remote supervision through phone calls and virtual check-ins further improved access to hard-to-reach areas.

3. *Performance Management (PM) Dashboard:* Real-time performance dashboards visualize key performance indicators (KPIs), enabling supervisors to track progress at the health post. The dashboard's analytical tools supported decision-making and allowed supervisors to recognize trends and address bottlenecks early.

***Developing eCHIS Dashboard***: The eCHIS dashboard provides supervisors and program managers with a unified view of HEW's performance within and across HPs and woredas. By displaying KPIs' targets against performance, the dashboard facilitated data-driven decisions. The dashboard's monthly performance tracking allows decision-makers to assess progress in selected indicators, ensuring targeted support and evidence-based actions. Decision-makers could pinpoint areas requiring intervention, enabling resource allocation to underperforming HPs and woredas. Real-time visualization of data trends empowered health administrators to respond proactively to service delivery gaps to make evidence-based decisions to maximize their impact.

Built using Apache Superset, the eCHIS dashboard is hosted at the Ministry of Health. Authorized users, including those at the health center level, can easily access their performance data against targets for selected indicators.

**B. Equipped:** Equipping the HEWs and supervisors with required inputs for optimal delivery of RMNCH and HEP services.

### Plan Setting Manager

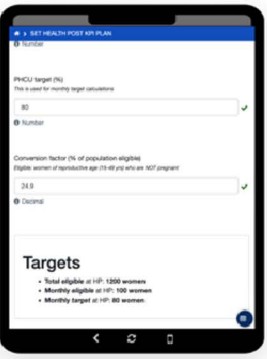

### PM Dashboard

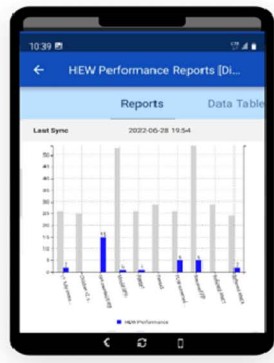

### Supervisor Task Manager

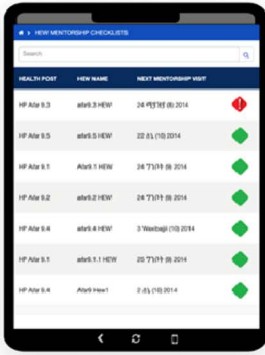

**Fig 1. Enhanced Focal Person Application (FPA) with Plan Setting Manager, PM Dashboard, and Supervisor Task Manager features.**

Integrating digital tools into routine workflows is a cornerstone of the intervention's success. The HEWs and supervisor should be equipped with a functioning tablet, eCHIS application, FPA, access to the internet, and a power source for charging to ensure routine use of the digital system and digital solutions as prescribed by the PM/PBI interventions. Moreover, reliable technical and troubleshooting supports require consideration for implementing the digitally enabled PM/PBI interventions with fidelity. Comprehensive training sessions equip HEWs, supervisors, and managers with the skills to use the eCHIS application, FPA, and eCHIS dashboard effectively. Training modules covered KPIs, service provision using eCHIS, performance targets, performance monitoring, evaluation and reporting, and decision-making processes. Refresher training can address emerging challenges, ensuring sustained competency and adherence to best practices. Inputs for RMNCH service provision, including the eight essential drugs and medical supplies specified in the national HMIS document that should be available all the time to ensure uninterrupted service delivery, are specially considered in the PM/PBI interventions.

**C. Supervised:** supervisors support the HEWs in improving their performance and achieving targets.

Supervisory checklists and real-time dashboards were integrated into the FPA to streamline support and monitoring processes. These tools empowered supervisors to provide tailored support to CHWs, addressing specific gaps in their performance. Additionally, the application provides supervisors with task manager support for follow-up actions, ensuring consistency in oversight. Supervisors use the FPA to monitor performance and identify specific areas requiring intervention. The PM dashboard provides comprehensive, real-time performance data, consolidating KPIs such as immunization coverage, growth monitoring, antenatal care visits, nutrition screening, family planning uptake, model households, supervision undertaking, and tracking essential RMNCH medicines and supplies. This dashboard empowered supervisors to make evidence-based decisions, improving support and service delivery. Regular performance reviews based on dashboard insights provide actionable insights to address gaps and recognize high-performing individuals and teams to sustain improvement.

**D. Compensated**: PBI introduces incentives to reward the attainment of results on selected KPIs to promote hard work, innovation, and accountability.

Incentives can be monetary or non-monetary. According to the HCD and co-design workshops, the HEWs gravitated towards non-financial incentives. These include in-service training, respect, recognition, opportunities for career growth, and in-kind items (smartphones, laptops, trophy cups, and kitchen utensils). As the success of any performance-based incentive scheme depends upon the verification of performance results, careful considerations should be made. Optimal utilization of the eCHIS application suite is critical to evaluate HEWs, supervisor's and HPs' performance accurately using digital data, which can foster objective measurement for recognizing desired performance.

The performance verification is to be undertaken in two steps. First, the PBI governance and management team at the HC verify the performance of the HEWs and HPs on the selected KPIs using data from the eCHIS application suite. The team will shortlist high-performing HEWs and HPs deserving the incentive per the agreed-upon evaluation criteria and performance target. This is followed by verification by the woreda PBI governance and management team, who have access to the national eCHIS dashboard that displays HEWs and supervisors' performance data against set targets. The verification teams can performance validations by taking random samples in the community. Verifying the performance of the supervisors is the responsibility of the Woreda governance and management team. Finally, the best performing HEW, supervisors, and HPs ranking in the top three will receive the incentives they are entitled to as agreed upon during the co-design workshop. The woreda organizes the award ceremony biannually. Award recipients compete with all their counterparts HEWs, HPs, and HCs in the woreda. Although HPs, HEWs, and Supervisors with average scores of KPIs above 70% are eligible for the incentive, only those who are in the top three ranks receive the incentive. The amount or the value of the incentive varies depending on the recipient's rank. Moreover, the kind and volume of items given as incentives can be determined by implementing partners, available budgets, and respective administrative bodies.

## Discussion

The study design demonstrated the importance of employing multiple participatory approaches in designing digitally enabled performance management interventions to ensure adoption, feasibility, scalability, and sustainability. This involves enhancing the FPA and developing the national eCHIS dashboard as core digital innovations, which can significantly enhance supervision, data management, and decision-making processes. These digital tools provide actionable insights on setting performance targets for KPIs, supervision, performance monitoring and feedback, and data-informed decisions. Leveraging the eCHIS application suite, the interventions integrated key components of the DESC framework, which proves its relevance for improving community health worker motivation, performance, accountability, and supervision. By weaving PM/PBI digital innovations into the fabric of the HEP rather than layering it on top, the design is intended to strengthen the HEW performance and motivation, improve RMNCH service provision, and accelerate progress toward universal health coverage.

The FPA's automation capabilities streamlined supervisory activities, including target setting, mentoring, supervision, and performance monitoring in real-time, and allow supervisors to provide timely feedback and support. By doing so, the FPA features not only reduce administrative burdens but also enhance the clarity of HEWs' responsibilities and KPIs, which are aligned with HEP priorities and RMNCH goals, fostering adoption, accountability, and improved service provision. This is supported by studies that demonstrated the benefit of setting clear expectations and enhancing support for improving accountability and service quality, and for making objectives relevant and impactful [12,17].

Consistent with evidence from several community health programs, the eCHIS national and the PM dashboards facilitate data visualization and analysis and hence can empower supervisors and decision-makers to identify gaps, and for improving motivation, performance, accountability, efficiency, and strategic decision [11,13,15,17]. Regular performance reviews, informed by dashboard insights, can provide actionable feedback to drive continuous improvement, minimize disruptions to existing practices, while maximizing the utility of the digital tools.

The sustainability and scalability of digital health interventions hinge on their ability to integrate into existing systems, secure local ownership, and adapt to evolving contexts [10,14].

A scoping review of comparative studies focusing on designing and piloting digital health interventions in low-resource settings pointed out major principles to foster adoption, sustainability, and scale up [10]. From the design perspective, the digitally enabled PM/PBI innovations are aligned closely with sustainability and scalability principles outlined in the scoping review [10]. *Ensuring institutional commitment and governance structures* for financial and operational sustainability is one of the important principles. In this regard, Ethiopia's Health Sector Transformation Plan II explicitly designates digital health as a strategic objective, and the Ministry of Health has spearheaded the rapid national rollout of DHIS2 and eCHIS. In addition, the HEP optimization roadmap recommends the introduction of innovative performance management systems, demonstrating the technical capacity and high-level buy-in [6,22]. Moreover, the focus on non-monetary incentives, such as career advancement, recognition, and in-kind rewards, reflects stakeholder preferences and aligns with Ethiopia's resource constraints. This is consistent with WHO recommendations for non-financial incentives, such as professional development opportunities and public recognition to motivate CHWs and reinforce their commitment to delivering high-quality care, which is supported by another study demonstrating the benefit of non-financial incentives in fostering intrinsic motivation among CHWs, leading to sustained improvements in performance [7,24].

*Stakeholder Ownership and Institutional Integration principle* relates to co-designing of the PM/PBI interventions, including the FPA, with national, regional, and district decision makers, supervisors, and HEWs, as well as insights gained from HCD. This approach mirrors the LIFE app's success in Kenya, where clinician input ensured relevance to training workflows. Moreover, the digitally enabled PM/PBI innovations are designed to be aligned with RMNCH and HEP goals, contextually tailored, and pragmatic for improving workforce motivation and performance in HEP service provision and RMNCH outcomes, which would ultimately drive HEP revitalization. Extensive stockholders' engagement made the design not only contextually relevant but also fostered a sense of ownership among stakeholders, which could enhance the sustainability and scalability of the interventions.

*Capacity building for long-term success principal* concerns with hidden costs of technical and troubleshooting support and maintenance, which often disrupt digital initiatives. The PM/PBI intervention addresses this by prioritizing local capacity development through competency-based training, troubleshooting, and refresher sessions for HEWs, supervisors, and managers. According to Mettler and Rohner, who wrote about Performance Management in health care, regular refresher sessions reinforce best practices and address emerging challenges, ensure sustained health workers' competency [12].

*Resource Alignment* is the other principle, while the PM/PBI innovations, including the national eCHIS dashboards and the FPA, enhance data-driven support and decision-making; their success depends on access to devices and logistics. JSI is a reliable partner for MoH working on operationalizing some strategies of the 2019 HEP roadmap (including the designing and testing of these digitally enabled PM/PBI innovations) [6] has provided high specification tablets, solar chargers, power banks and local wi-fi hotspots to ensure successful pilot implementation of the digital PM/PBI innovations. This demonstrated that even in low-resource settings like Ethiopia, judicious investment in hardware and connectivity by involving partners can render digitized service provision, real-time dashboards, and automated supervisory support and monitoring feasible. Similarly, Tanzania's mHealth job-aid for family-planning CHWs overcame connectivity and device-charging barriers through partnerships with local telecoms and provision of solar chargers, demonstrating the value of infrastructure partnerships in similar rural contexts [10]. Nonetheless, ongoing reliance on donor financing and partner support for device procurement and software maintenance could be a risk and hence calls for consideration of domestic financing. The digitally enabled PBI innovation focuses on non-monetary incentives (e.g., recognition), aligning with Text-MATCH in New Zealand, which tailors content to cultural preferences [10].

*Pragmatic adaptation and financial security* principles relate to the digitally enabled PM/PBI innovations' iterative design resonating with Tanzania's mobile job aid, which underwent multiple adaptations before scaling. Moreover, by integrating the FPA into HEP supervisory workflows and linking KPIs to national targets, the PM/PBI innovations position themselves for institutionalization and domestic funding, similar to New Zealand's TextMATCH, sustained through district health budgets [10]. South Africa's MomConnect SMS platform similarly highlighted that despite early success in scaling registrations, weak budgetary commitment at provincial levels threatened sustainability [25].

*Evaluation for Iterative Learning* is the other important principle to be considered during the piloting/testing of the digitally enabled PM/PBI interventions. According to a trial on health intervention in New Zealand, co-design alone did not guarantee behavior change, underscoring the need for context-specific evaluations [10]. In our case, consideration has been made to conduct systemic and robust evaluation of the digitally enabled PM/PBI pilot testing employing Implementation research approach to examine the reach, effectiveness, adoption, implementation (feasibility, fidelity and acceptability), sustainability and to explore barriers that hinder and enablers that facilitate real world implementation of the interventions.

**Lessons Learned**: The success of the design process highlighted the importance of combining digital innovations with participatory design to effectively address systemic challenges with the performance management systems. Stakeholder engagement at every stage ensured that the interventions were practical, culturally appropriate, aligned with national health goals, and scalable for community health programs in similar settings. The lessons learned offer valuable insights for similar initiatives aiming to enhance CHW performance and achieve sustainable health improvements. Furthermore, the integration of non-financial incentives underscored their value in CHW motivation and performance as well as scalability, leveraging existing health system resources.

## Conclusion

In conclusion, the digitally enabled performance management interventions designed for Ethiopia's HEP can represent a feasible, scalable, and sustainable model for improving CHW motivation, performance, and RMNCH outcomes in similar contexts. By leveraging digital tools, fostering collaboration, and prioritizing stakeholder engagement, the design addressed systemic challenges and demonstrated its potential to achieve long-term impact. Future efforts should focus on testing

these innovations and documenting the learning to advise on scaling up the interventions. The implementation of the PM/PBI interventions can be affected by the inaccessibility of internet connectivity, power sources for charging the digital tools, the digital tool performance capacity to run the eCHIS application suite and dashboard use, and financial constraints. These are foreseen challenges requiring attention when intending to implement the PM/PBI interventions with fidelity. Additionally, frequent updates through training to navigate the digital applications and tools effectively are an important consideration. To address these challenges, ongoing investments in infrastructure, capacity building, budget allocation, and reliable technical support are needed. Studies are needed to test the interventions in improving motivation, performance, service provision, and RMNCH outcomes and evaluate the long-term effects of these interventions on health outcomes and system efficiency.

## Supporting information

**S1 Table.  Key Performance Indicators (KPIs) for measuring the performance of HEWs, HPs and supervisors.** (DOCX)

## Author contributions

**Conceptualization:** Alemnesh Hailemariam Mirkuzie, Yared Kifle, Gizachew Tadele Tiruneh, Girma Tadesse, Getnet Alem Teklu, Esubalew Sebsibe, Eyoel Mitiku, Aklilu Abera, Wondwossen Shiferaw, Birhutesfa Bekele, Wuleta Aklilu Betemariam, Desalew Emaway.

**Data curation:** Alemnesh Hailemariam Mirkuzie, Yared Kifle.

**Formal analysis:** Alemnesh Hailemariam Mirkuzie.

**Funding acquisition:** Gizachew Tadele Tiruneh, Birhutesfa Bekele, Wuleta Aklilu Betemariam, Desalew Emaway.

**Investigation:** Alemnesh Hailemariam Mirkuzie, Getnet Alem Teklu, Esubalew Sebsibe, Aklilu Abera, Wondwossen Shiferaw.

**Methodology:** Alemnesh Hailemariam Mirkuzie, Yared Kifle, Gizachew Tadele Tiruneh, Girma Tadesse, Getnet Alem Teklu, Wondwossen Shiferaw, Birhutesfa Bekele.

**Project administration:** Girma Tadesse, Getnet Alem Teklu, Esubalew Sebsibe, Aklilu Abera, Wondwossen Shiferaw, Desalew Emaway.

**Resources:** Girma Tadesse, Eyoel Mitiku, Wuleta Aklilu Betemariam, Desalew Emaway.

**Software:** Eyoel Mitiku.

**Supervision:** Alemnesh Hailemariam Mirkuzie, Yared Kifle, Girma Tadesse, Getnet Alem Teklu, Esubalew Sebsibe, Aklilu Abera, Wondwossen Shiferaw, Birhutesfa Bekele, Desalew Emaway.

**Validation:** Alemnesh Hailemariam Mirkuzie, Yared Kifle, Gizachew Tadele Tiruneh, Esubalew Sebsibe, Eyoel Mitiku.

**Visualization:** Alemnesh Hailemariam Mirkuzie, Gizachew Tadele Tiruneh, Eyoel Mitiku, Wondwossen Shiferaw.

**Writing – original draft:** Alemnesh Hailemariam Mirkuzie, Yared Kifle, Gizachew Tadele Tiruneh, Getnet Alem Teklu, Esubalew Sebsibe, Eyoel Mitiku, Aklilu Abera, Wondwossen Shiferaw, Wuleta Aklilu Betemariam, Desalew Emaway.

**Writing – review & editing:** Alemnesh Hailemariam Mirkuzie, Yared Kifle, Gizachew Tadele Tiruneh, Girma Tadesse, Getnet Alem Teklu, Esubalew Sebsibe, Eyoel Mitiku, Aklilu Abera, Wondwossen Shiferaw, Birhutesfa Bekele, Wuleta Aklilu Betemariam, Desalew Emaway.

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
