## [Decision Letter · Decision Letter 0]

PDIG-D-25-00007What does it take to design digitally enabled performance management and incentive interventions for community health programs: Lessons from EthiopiaPLOS Digital Health Dear Dr. Mirkuzie, Thank you for submitting your manuscript to PLOS Digital Health. After careful consideration, we feel that it has merit but does not fully meet PLOS Digital Health's publication criteria as it currently stands. Therefore, we invite you to submit a revised version of the manuscript that addresses the points raised during the review process. Please submit your revised manuscript within 60 days May 11 2025 11:59PM. If you will need more time than this to complete your revisions, please reply to this message or contact the journal office at digitalhealth@plos.org. Please include the following items when submitting your revised manuscript:* A rebuttal letter that responds to each point raised by the editor and reviewer(s). You should upload this letter as a separate file labeled 'Response to Reviewers '. This file does not need to include responses to any formatting updates and technical items listed in the 'Journal Requirements' section below.* A marked-up copy of your manuscript that highlights changes made to the original version. You should upload this as a separate file labeled 'Revised Manuscript with Track Changes '.* An unmarked version of your revised paper without tracked changes. You should upload this as a separate file labeled 'Manuscript '. If you would like to make changes to your financial disclosure, competing interests statement, or data availability statement, please make these updates within the submission form at the time of resubmission. Guidelines for resubmitting your figure files are available below the reviewer comments at the end of this letter. We look forward to receiving your revised manuscript. Kind regards, Laura Sbaffi, PhD, MA, MScSection EditorPLOS Digital Health Kiran PaudelAcademic EditorPLOS Digital Health Leo Anthony CeliEditor-in-ChiefPLOS Digital Healthorcid.org/0000-0001-6712-6626 **Journal Requirements:**

2. We have amended your Competing Interest statement to comply with journal style. We kindly ask that you double check the statement and let us know if anything is incorrect. **Additional Editor Comments (if provided):****Reviewers' Comments:** Reviewer's Responses to Questions

**Comments to the Author**

1. Does this manuscript meet PLOS Digital Health’s publication criteria ? Is the manuscript technically sound, and do the data support the conclusions? The manuscript must describe methodologically and ethically rigorous research with conclusions that are appropriately drawn based on the data presented.

Reviewer #1: Partly

2. Has the statistical analysis been performed appropriately and rigorously?

Reviewer #1: I don't know

3. Have the authors made all data underlying the findings in their manuscript fully available (please refer to the Data Availability Statement at the start of the manuscript PDF file)?

Reviewer #1: Yes

4. Is the manuscript presented in an intelligible fashion and written in standard English?

Reviewer #1: No

5. Review Comments to the Author

Reviewer #1: 1.Introduction: The problem statement is strong but could include more references to previous digital health PM/PBI interventions in similar settings.

2. Methodology Section: Needs more clarity and rigor. The study lacks explicit methodological details about study participants and data collection methods (on how data was collected, analyzed, and validated).

3. Results & Discussion: Reduce redundancy—many points in the discussion are already covered in the results.

Add a deeper analysis of challenges and feasibility. Discuss how the Ethiopian context may influence the scalability of this model. Add a comparative discussion with similar digital health interventions from other low-resource settings.

4.Conduct a thorough grammatical review and restructuring to remove redundancy, grammatical errors and improve clarity.

6. PLOS authors have the option to publish the peer review history of their article (what does this mean? ). If published, this will include your full peer review and any attached files.

**Do you want your identity to be public for this peer review?** For information about this choice, including consent withdrawal, please see our Privacy Policy .

Reviewer #1: No

---

## [Editor Report · Decision Letter 1]

What does it take to design digitally enabled performance management and incentive interventions for community health programs: Lessons from Ethiopia

PDIG-D-25-00007R1

Dear Dr. Mirkuzie,

We are pleased to inform you that your manuscript 'What does it take to design digitally enabled performance management and incentive interventions for community health programs: Lessons from Ethiopia' has been provisionally accepted for publication in PLOS Digital Health.

Best regards,

Laura Sbaffi, PhD, MA, MSc

Section Editor

PLOS Digital Health